# Engineering ER-stress dependent non-conventional mRNA splicing

**Weihan Li[1,2]\*, Voytek Okreglak[1,2]†, Jirka Peschek[1,2], Philipp Kimmig[1,2]‡, Meghan Zubradt[2,3], Jonathan S Weissman[2,3], Peter Walter[1,2]\***

[1]Department of Biochemistry and Biophysics, University of California, San Francisco, San Francisco, United States; [2]Howard Hughes Medical Institute, San Francisco, United States; [3]Department of Cellular and Molecular Pharmacology, University of California, San Francisco, San Francisco, United States

**Abstract** The endoplasmic reticulum (ER) protein folding capacity is balanced with the protein folding burden to prevent accumulation of un- or misfolded proteins. The ER membrane-resident kinase/RNase Ire1 maintains ER protein homeostasis through two fundamentally distinct processes. First, Ire1 can initiate a transcriptional response through a non-conventional mRNA splicing reaction to increase the ER folding capacity. Second, Ire1 can decrease the ER folding burden through selective mRNA decay. In *Saccharomyces cerevisiae* and *Schizosaccharomyces pombe,* the two Ire1 functions have been evolutionarily separated. Here, we show that the respective Ire1 orthologs have become specialized for their functional outputs by divergence of their RNase specificities. In addition, RNA structural features separate the splicing substrates from the decay substrates. Using these insights, we engineered an *S. pombe* Ire1 cleavage substrate into a splicing substrate, which confers *S. pombe* with both Ire1 functional outputs.
DOI: https://doi.org/10.7554/eLife.35388.001

**\*For correspondence:**
weihan@walterlab.ucsf.edu (WL);
peter@walterlab.ucsf.edu (PW)

**Present address:** †Calico Life Sciences LLC, South San Francisco, United States; ‡Institute of Biochemistry, ETH Zürich, Zürich, Switzerland

**Competing interests:** The authors declare that no competing interests exist.

## Introduction

In eukaryotes, the vast majority of secretory and transmembrane proteins are folded in the endoplasmic reticulum (ER). The ER protein folding homeostasis is maintained by a collective of signaling pathways, termed the unfolded protein response (UPR) (*Walter and Ron, 2011*; *Ron and Walter, 2007*). The most evolutionarily conserved branch of the UPR is mediated by the ER-transmembrane kinase/endoribonuclease (RNase) Ire1. Direct binding of unfolded proteins to Ire1's ER lumenal domain triggers Ire1 to oligomerize and form foci (*Gardner and Walter, 2011*; *Karagöz et al., 2017*; *Credle et al., 2005*; *Aragón et al., 2009*). In turn, Ire1 oligomerization activates Ire1's cytosolic kinase/RNase domain (*Korennykh et al., 2009*), which restores ER homeostasis through two functional outputs. First, Ire1 initiates a process of non-conventional cytosolic splicing of *XBP1* mRNA (in metazoans) or *HAC1* mRNA (in *S. cerevisiae*). Translation of the spliced mRNA produces a transcription factor Xbp1 (Hac1 in *S. cerevisiae*), which drives a large transcriptional program to adjust the ER's protein-folding capacity according to the protein folding load in the ER lumen (*Cox et al., 1993*; *Mori et al., 1993*; *Yoshida et al., 2001*; *Calfon et al., 2002*; *Sidrauski et al., 1996*). Second, Ire1 can reduce the ER folding burden by cleaving a set of mRNAs encoding ER-target proteins. The initial Ire1-mediated cleavage leads to mRNA degradation, in a process termed regulated Ire1-dependent decay (RIDD) (*Hollien and Weissman, 2006*; *Hollien et al., 2009*; *Kimmig et al., 2012*). The mechanism that distinguishes the non-conventional mRNA splicing from RIDD has largely remained unknown.

Interestingly, the two Ire1 modalities co-exist in metazoan cells (*Hollien and Weissman, 2006*; *Hollien et al., 2009*; *Moore and Hollien, 2015*), yet are evolutionarily separated in the two yeast species, *S. cerevisiae* and *S. pombe*. The UPR in *S. cerevisiae* engages Ire1 exclusively in mRNA

splicing, whereas in *S. pombe* it engages Ire1 exclusively in RIDD. There is no detectable RIDD in *S. cerevisiae* and no *HAC1/XBP1* ortholog in *S. pombe*, nor is there a corresponding transcriptional program (*Niwa et al., 2005*; *Kimmig et al., 2012*). It is intriguing to note that the fundamental task of maintaining ER protein homeostasis can be achieved by two radically different processes catalyzed by two distantly related Ire1 orthologs. The two yeast species, *S. cerevisiae* and *S. pombe*, therefore provide a unique opportunity to dissect the two Ire1 functional outputs, which has remained an unsolved challenge in metazoans. Here, we set out to exploit this opportunity.

## Results

### *S. pombe* and *S. cerevisiae* Ire1's cytosolic domains are functionally divergent

Despite their common role as UPR effectors and conserved domain structures (*Figure 1A*), *S. pombe* and *S. cerevisiae* Ire1 orthologs share 29% sequence identity, and sequence variation may confer Ire1's functional divergence. To explore this notion, we swapped the homologous *IRE1* genes between the two yeast species by expressing *S. cerevisiae* Ire1 in Δ*ire1 S. pombe* cells and, *vice versa*, *S. pombe* Ire1 in Δ*ire1 S. cerevisiae* cells. To this end, we constructed strains in which we integrated the foreign *IRE1* genes into the genomes of the other yeast such that their expression was regulated by the host species-endogenous *IRE1* promoters and the resulting mRNAs contained host species-endogenous 5' and 3' untranslated regions (UTR). The *IRE1* genes contained sequences encoding FLAG-tags that we inserted into an unstructured loop in their ER-lumenal domains in a position known to preserve Ire1 function (*Rubio et al., 2011*). In both yeasts, the foreign genes expressed Ire1 at comparable levels (*Figure 1B and C*, lanes 3). However, when grown on plates containing tunicamycin, a drug that blocks N-linked glycosylation and induces ER stress, the foreign Ire1s failed to support cell growth of either *S. pombe* and *S. cerevisiae* cells (*Figure 1D and E*, lanes 3), indicating that *S. pombe* and *S. cerevisiae* Ire1s are not interchangeable.

There are two plausible, not mutually exclusive scenarios that could explain the failure of cross-species complementation: (i) the foreign Ire1 lumenal domains fail to sense ER stress, or (ii) the foreign Ire1 cytosolic domains fail to recognize species-appropriate RNA substrates. Since the Ire1 lumenal domains have lower sequence identity (21%) than the cytosolic kinase/RNase domains (45%), we first swapped the Ire1 lumenal domains, generating chimeras with foreign lumenal domains and host species-endogenous transmembrane/cytosolic domains. Both chimeras supported growth on tunicamycin plates, suggesting that the divergent Ire1 lumenal domains share a conserved mechanism to sense ER stress and transduce the signal across ER membrane (*Figure 1D and E*, lanes 4). Next, we swapped the Ire1 transmembrane/cytosolic domains. These Ire1 chimeras failed to restore growth on tunicamycin plates of both *S. pombe* and *S. cerevisiae* cells (*Figure 1D and E*, lanes 5), indicating that the Ire1 transmembrane/cytosolic domains cause the Ire1 functional incompatibility when expressed in the opposing yeast. As a control, we expressed FLAG-tagged host species-endogenous Ire1s into Δ*ire1* strains of both yeasts. These strains phenocopied the wild type (WT) cells on tunicamycin plates (*Figure 1D and E*, lanes 6). We again confirmed by immunoblotting that all of the FLAG-tagged Ire1 constructs were stably expressed at near-endogenous level in both yeasts (*Figure 1B and C*).

We next asked whether the Ire1 constructs would process the host species-appropriate RNA substrates in *S. pombe* and *S. cerevisiae* cells. To this end, we performed Northern blot and qPCR analyses to measure cleavage and subsequent down-regulation of *GAS2* mRNA, which is a RIDD target in *S. pombe* cells (*Kimmig et al., 2012*). We performed the Northern blot in Δ*ski2 S. pombe*, in which the RNA 3' to 5' decay machinery is impaired so that the *GAS2* mRNA 5' cleavage fragments can be detected in the gel. Of the different Ire1 variants, only the Ire1 chimera bearing the *S. pombe* cytosolic domain cleaved the *GAS2* mRNA (*Figure 1—figure supplement 1A*) and decreased the mRNA level (*Figure 1F*), consistent with the growth phenotype. In *S. pombe*, Ire1 also cleaves the *BIP1* mRNA within its 3'UTR, producing a truncated mRNA with an increased half-life (*Kimmig et al., 2012*). To assess *BIP1* mRNA processing, we performed qPCR analysis using two pairs of primers, one pair bracketing the Ire1 cleavage site and the other pair bracketing a region upstream of it (*Figure 1G*, schematic insert, *black* vs. *grey* arrows), reporting on uncleaved only and both *BIP1* mRNA species (i.e. total *BIP1* mRNA), respectively. As expected, upon tunicamycin-induced ER

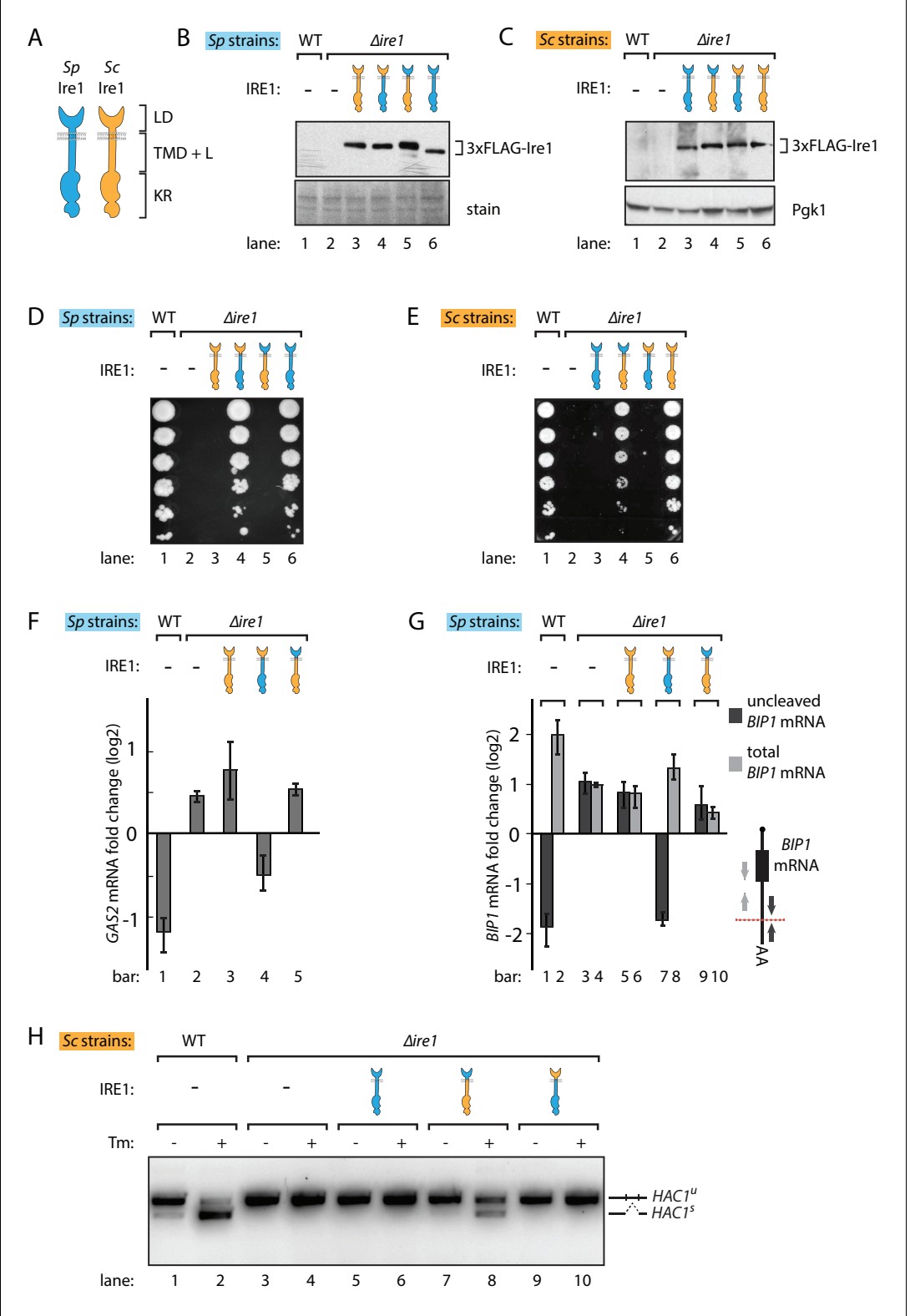

**Figure 1.** *S. pombe* and *S. cerevisiae* Ire1 have functionally conserved stress sensing ER-lumenal domains and divergent cytosolic domains. (**A**) Cartoon illustration of lumenal domain (LD), transmembrane/cytosolic linker domain (TMD + L) and kinase/RNase domain (KR) for *S. pombe* (*Sp*) (*blue*) and *S. cerevisiae* (*Sc*) Ire1 (*orange*). (**B, C**) Expression levels of *S. cerevisiae* Ire1 (128 kD), *S. cerevisiae* lumenal *S. pombe* cytosolic Ire1 (126 kD), *S. pombe* lumenal *S. cerevisiae* cytosolic Ire1 (125 kD) and *S. pombe* Ire1 (122 kD) in *S. pombe* (**B**) and *S. cerevisiae* cells (**C**).
*Figure 1 continued on next page*

*Figure 1 continued*

Extracts were immunoblotted for 3xFLAG-Ire1. Ponceau stain (**B**) or Pgk1 (**C**) was used as loading control. (**D, E**) Cell growth assay on tunicamycin (Tm) plates. Serial dilutions of *S. pombe* (**D**) or *S. cerevisiae* (**E**) cells, which expressed the indicated Ire1 constructs, were spotted onto plates containing 0.05 µg/ml (**D**) or 0.1 µg/ml (**E**) of Tm. Plates were photographed after incubation at 30°C for 4 days. (**F, G**) qPCR assay for *S. pombe GAS2* (**F**) or *BIP1* (**G**) mRNA fold change upon 1 µg/ml Tm treatment for 1 hr. Experiments were done in triplicates. In (**G**), uncleaved (*dark grey*) or total (*light grey*) *BIP1* mRNA was detected using the corresponding PCR primers illustrated as arrows in the schematic insert. The red dashed line indicates the Ire1 cleavage position on *BIP1* mRNA. (**H**) Detection of *S. cerevisiae HAC1* mRNA splicing by RT-PCR across the splice junction. Cells were treated with or without 1 µg/ml of Tm for 1 hr.

DOI: https://doi.org/10.7554/eLife.35388.002

The following figure supplements are available for figure 1:

**Figure supplement 1.** Ire1 chimeras with *S. pombe* cytosolic domain cleave *BIP1* and *GAS2* mRNA in *S. pombe*.
DOI: https://doi.org/10.7554/eLife.35388.003

**Figure supplement 2.** Ire1 oligomeric state determines the *HAC1* mRNA splicing dynamics in *S. cerevisiae* cells.
DOI: https://doi.org/10.7554/eLife.35388.004

stress in WT cells uncleaved *BIP1* mRNA levels decreased while total *BIP1* mRNA level increased (*Figure 1G*, lanes 1, 2). As shown in *Figure 1G*, Ire1 variants bearing the *S. pombe* cytosolic domain processed *BIP1* mRNA, whereas Ire1 variants bearing the *S. cerevisiae* cytosolic domain did not. This result was further validated by Northern blot analysis of *BIP1* mRNA (*Figure 1—figure supplement 1B*).

Correspondingly in *S. cerevisiae* cells, we examined *HAC1* mRNA splicing by PCR across its splice junction. Consistent with the cell growth phenotype on tunicamycin, the two Ire1 constructs bearing the *S. pombe* cytosolic domains did not splice *HAC1* mRNA in *S. cerevisiae* cells (*Figure 1H*, lanes 6, 10). By contrast, the Ire1 chimera bearing the *S. pombe* lumenal domain and *S. cerevisiae* cytosolic domains spliced *HAC1* mRNA (*Figure 1H*, lane 8), albeit at reduced efficiency compared to WT *S. cerevisiae* Ire1 (*Figure 1H*, lane 2). We confirmed the activity of the various Ire1 constructs in *HAC1* mRNA splicing by monitoring UPR dynamics with a *HAC1* mRNA-derived splicing reporter (*Figure 1—figure supplement 2A and B*) previously described (*Aragón et al., 2009*; *Zuleta et al., 2014*). The reduced *HAC1* mRNA splicing efficiency observed for Ire1 bearing the *S. pombe* lumenal domain (*Figure 1H*, lane 8, and *Figure 1—figure supplement 2B*, column 4) can be explained by the observation that the *S. pombe* lumenal domain mediates a lower degree of oligomerization than its *S. cerevisiae* counterpart, as demonstrated by the reduced ability of Ire1-mCherry fusion constructs to form foci visible by fluorescent microscopy (*Figure 1—figure supplement 2C*). Consistent with previous studies (*Aragón et al., 2009*), the insertion of the mCherry module into the Ire1 cytosolic linker, which connects Ire1 transmembrane domain and cytosolic kinase/RNase domain, did not affect its ability to sustain cell growth (*Figure 1—figure supplement 2D*).

## Ire1 kinase/RNase domains have distinct RNase specificity

To further confine the Ire1 region giving rise of the species differences in outputs, we expressed an Ire1 chimera that, in addition to the *S. cerevisiae* lumenal domain, also included the *S. cerevisiae* transmembrane and cytosolic linker domains fused to the *S. pombe* kinase/RNase domain. This chimeric Ire1 weakly rescued cell growth and mildly restored the *HAC1* mRNA splicing upon ER stress (*Figure 2A* lane 4, and *Figure 2B* lane 6), compared to the chimera containing *S. pombe* transmembrane and cytosolic linker domains, although both constructs were expressed at similar protein levels (*Figure 2C*). This result indicates that the major difference lies in the kinase/RNase domains, but that the transmembrane and cytosolic linker domains can afford a marginal rescue, most likely by reintroducing cytosolic linker elements that facilitate *HAC1* mRNA docking (*van Anken et al., 2014*).

To study the differences by which the Ire1 kinase/RNase domains select their respective substrate mRNAs, we purified recombinant *S. cerevisiae* and *S. pombe* kinase/RNase domains and performed in vitro RNA cleavage assays. The *S. cerevisiae* Ire1 kinase/RNase efficiently cleaved a cognate 29-nucleotide RNA hairpin derived from the 3' splice site of *S. cerevisiae HAC1* mRNA (*Figure 2D*). By contrast, under the same conditions the *S. pombe* Ire1 kinase/RNase cleaved the *S. cerevisiae HAC1* mRNA-derived substrate ~60 fold slower (*Figure 2D*). Reciprocally, the *S. pombe* Ire1 kinase/RNase

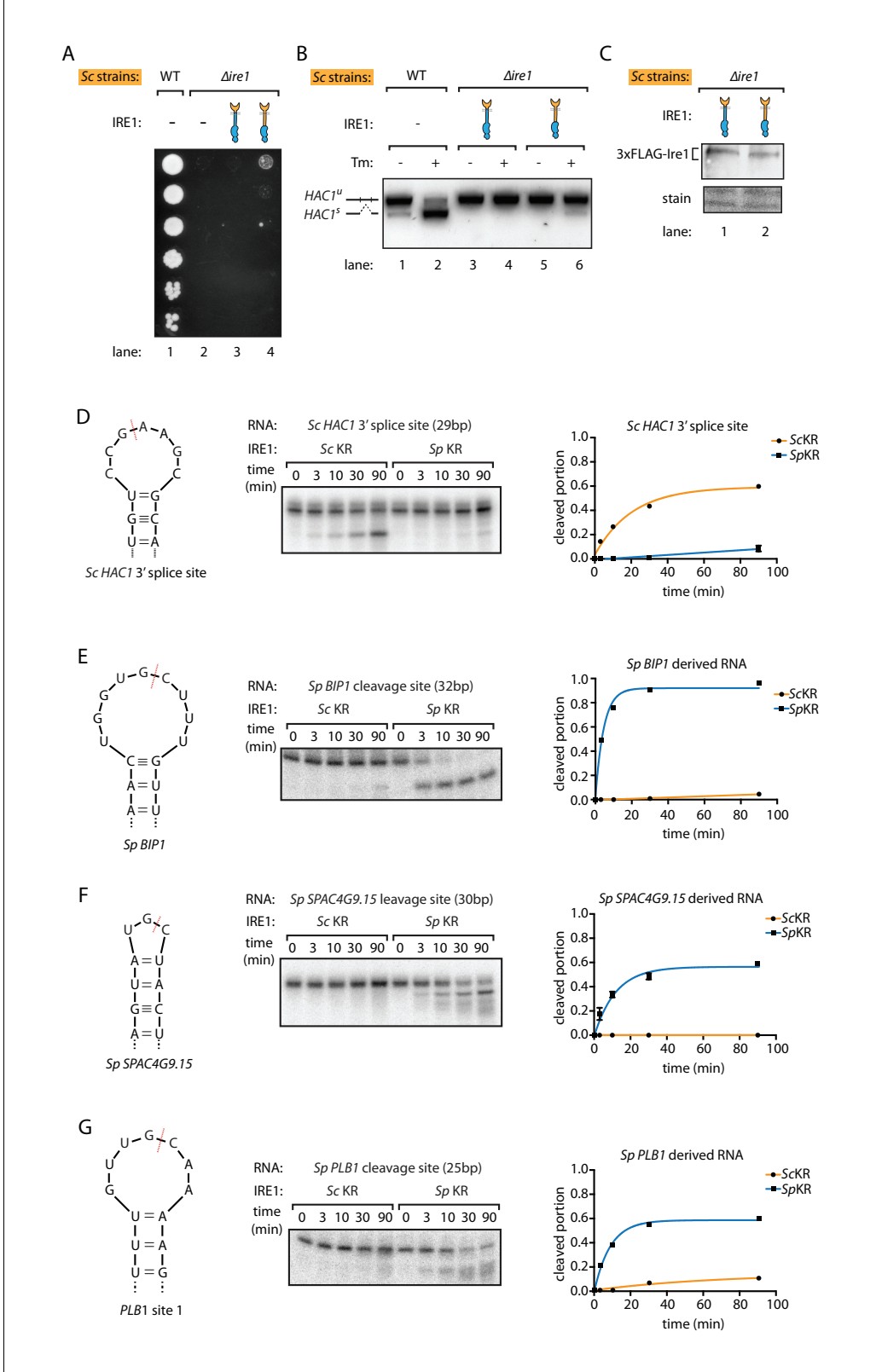

**Figure 2.** *S. pombe* and *S. cerevisiae* Ire1 have distinct RNase specificity. (**A**) Growth assay for *S. cerevisiae* cells expressing indicated Ire1 constructs on Tm plates, as *Figure 1E*. (**B**) Measuring *HAC1* mRNA splicing, as *Figure 1H*. (**C**) Comparing the expression levels of the indicated 3xFLAG-tagged Ire1 chimeras using immunoblotting. Ponceau stain was used as loading control. (**D, E, F, G**) In vitro RNA cleavage assays. 5'-radiolabeled hairpin RNA substrates were incubated with 12.5 µM *S. cerevisiae* or *S. pombe* Ire1 kinase/RNase domains (KR) at 30°C for the indicated time. (**D**) Hairpin RNA

*Figure 2 continued on next page*

*Figure 2 continued*

substrate derived from the 3' splice site of *S. cerevisiae HAC1* mRNA. The calculated $k_{obs}$ is $9.4 \pm 0.9 \times 10^{-4}$ s$^{-1}$ for *S. cerevisiae* Ire1 KR and $0.15 \pm 0.01 \times 10^{-4}$ s$^{-1}$ for *S. pombe* Ire1 KR. (E) Hairpin RNA substrate derived from the Ire1 cleavage site on *S. pombe BIP1* mRNA. The calculated $k_{obs}$ is $0.079 \pm 0.0006 \times 10^{-4}$ s$^{-1}$ for *S. cerevisiae* Ire1 KR and $37.3 \pm 4.4 \times 10^{-4}$ s$^{-1}$ for *S. pombe* Ire1 KR. (F) Hairpin RNA substrate derived from the Ire1 cleavage site on *S. pombe SPAC4G9.15* mRNA, encoding a gene of unknown function. The calculated $k_{obs}$ was below our detection limit for *S. cerevisiae* Ire1 KR and $15.6 \pm 2.2 \times 10^{-4}$ s$^{-1}$ for *S. pombe* Ire1 KR. (G) Hairpin RNA substrate derived from the Ire1 cleavage site on *S. pombe PLB1* mRNA. The calculated $k_{obs}$ is $0.2 \pm 0.003 \times 10^{-4}$ s$^{-1}$ for *S. cerevisiae* Ire1 KR and $19.0 \pm 2.5 \times 10^{-4}$ s$^{-1}$ for *S. pombe* Ire1 KR.

DOI: https://doi.org/10.7554/eLife.35388.005

efficiently cleaved a cognate 32-nucleotide RNA hairpin derived from the cleavage site in *S. pombe BIP1* mRNA and cognate RNA hairpins derived from the RIDD cleavage sites in *S. pombe SPAC4G9.15* and *PLB1*, whereas *S. cerevisiae* Ire1 kinase/RNase cleaved the *BIP1* mRNA-derived substrate and the *S. pombe SPAC4G9.15* >500 fold slower and *PLB1* mRNA-derived hairpins ~100 fold slower (*Figure 2E,F,G*). These in vitro data validate and expand the conclusions from the experiments conducted in vivo, suggesting that the different Ire1 RNase specificities separate their functional outputs and that they are not dependent on other cellular factors such as associated proteins or lipids.

## Ire1 kinase/RNase domains recognize distinct RNA sequence and structural features

Ire1 recognizes its substrates based on both RNA sequence and structural features. The required RNA sequence motifs were characterized previously and differ between species: for *S. cerevisiae* Ire1 a nucleotide sequence of CNG|CNGN or CNG|ANGN ('|' indicates the Ire1 cleavage position) situated in a strictly conserved 7-membered loop is required (*Gonzalez et al., 1999*). By contrast, for *S. pombe* Ire1 a three-nucleotide sequence of UG|C is required and no additional structural features have yet been characterized (*Kimmig et al., 2012*; *Guydosh et al., 2017*). To fill this gap in our knowledge, we examined the RNA secondary structures in vivo. To this end, we treated the *S. pombe* cells with dimethyl sulfate (DMS), which allows detection of exposed (unpaired and not blocked by proteins) adenine/cytosine residues (illustrated as green dots in *Figure 3A*). RNA was extracted, reverse transcribed and deep-sequenced. The DMS modifications stop reverse transcriptase and generate truncated DNA fragments that we mapped through deep sequencing. We then used identified unpaired bases to guide in silico RNA secondary structure predictions (*Rouskin et al., 2014*). For example, near one of the *GAS2* mRNA cleavage sites, five bases, labeled in green in *Figure 3B*, have high DMS modification signals. In the RNA folding software mfold (*Zuker, 2003*), we provided the constraint such that these five residues are unpaired and obtained the predicted RNA secondary structure (*Figure 3C*). In this structure, the *GAS2* mRNA forms a 9-membered stem loop with the cleavage consensus sequence UG|C located near the center of the loop. In similar analyses of 13 additional *S. pombe* Ire1 substrate mRNA cleavage sites previously identified by both Kimmig, Diaz et al. and Guydosh et al. (*Guydosh et al., 2017*; *Kimmig et al., 2012*), we found in all of them cleavage sites located near the center of loops in RNA stem-loop structures (*Figure 3—figure supplement 1A,B*). By contrast to those found in *S. cerevisiae HAC1* mRNA, the predicted loops were of variable sizes, with the smallest being a 3-membered loop (e.g., the *SPAC4G9.15* mRNA cleavage site) and the largest being a 9-membered loop (e.g., the *BIP1* mRNA cleavage site). We summarize that *S. pombe* Ire1 is tolerant to loop size variation, while the *S. cerevisiae* Ire1 stringently recognizes 7-membered stem loops. Thus, the *S. cerevisiae* and *S. pombe* Ire1 recognize distinct RNA sequence and structural features (*Figure 3D*).

A prediction of this model is that RNAs that combine *S. cerevisiae* and *S. pombe* Ire1 motifs should be substrates to Ire1 from either species. To test this prediction, we analyzed a substrate satisfying criteria for both species. Specifically, we examined a substrate predicted to form a 7-membered stem loop with the sequence CUG|CAGC, meeting the criteria of both the *S. cerevisiae* Ire1 motif (CNG|CNGN) and the *S. pombe* Ire1 motif (UG|C). Indeed, both enzymes cleaved this RNA in vitro with similar efficiency, in strong support of our model (*Figure 3E*).

We further challenged the model in vivo by modifying the *S. pombe BIP1* mRNA. First, we replaced the Ire1 cleavage site in the 3' UTR of *BIP1* mRNA with a sequence derived from the *S. cerevisiae HAC1* mRNA 5' splice site. This modification is predicted to change the endogenous 9-

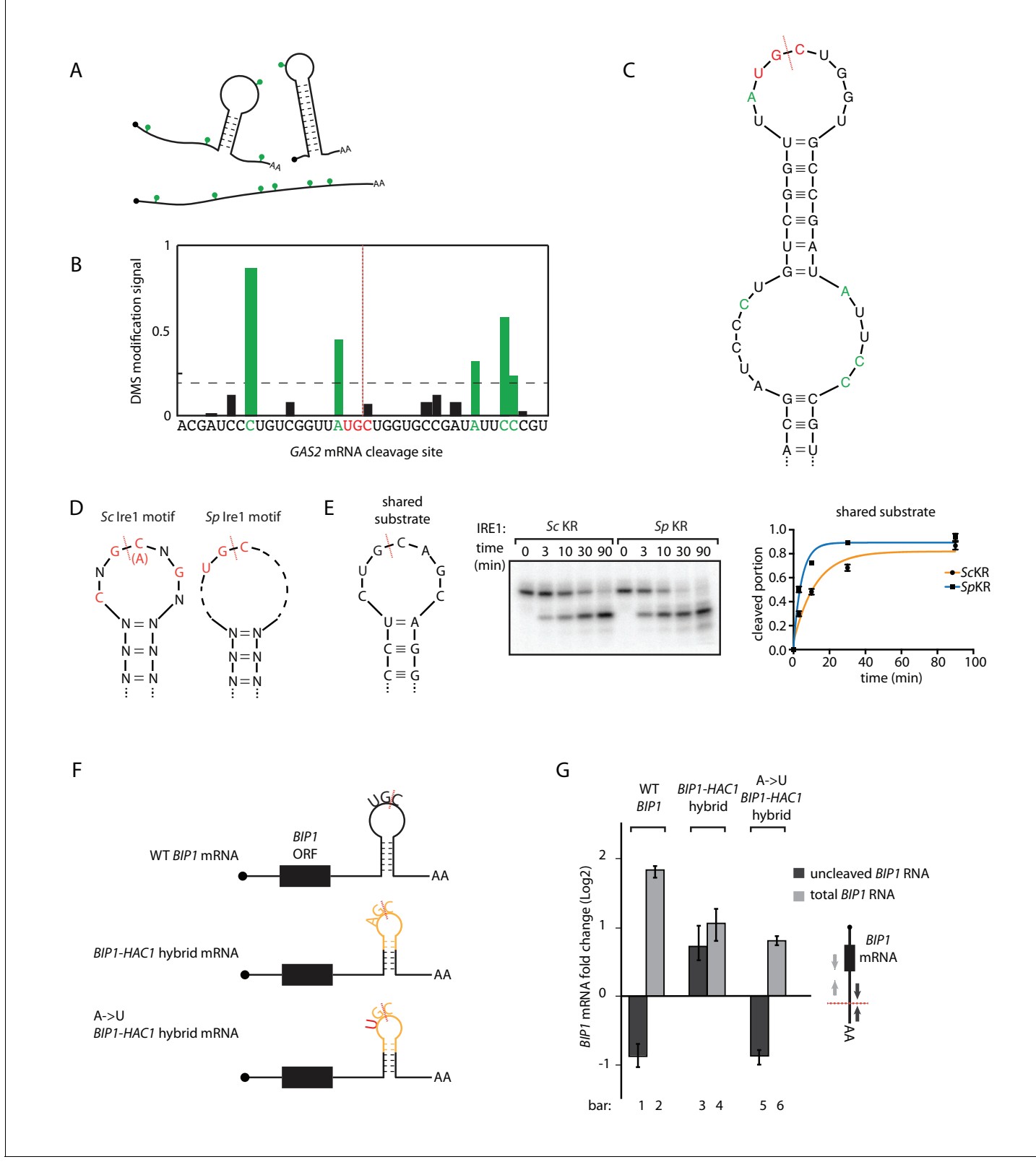

**Figure 3.** S.*pombe* and *S. cerevisiae* Ire1 recognize distinct RNA sequence and structural features. (**A**) Illustration of RNA structural mapping by DMS modifications. Dimethyl sulfate (DMS) allows detection of unpaired adenine and cytosine RNA bases (*green* dots). (**B**) The normalized DMS modification signals near the Ire1 cleavage site on *S. pombe GAS2* mRNA (cleavage site is indicated by the *red* dashed line). The positions with high DMS modification signals are labeled in *green* and the previously identified *S. pombe* Ire1 UG|C motif is labeled in *red*. (**C**) *In sillico* RNA secondary structure

*Figure 3 continued on next page*

*Figure 3 continued*

prediction of the Ire1 cleavage site on *GAS2* mRNA. Structure prediction was constrained by forcing the positions with high DMS modification signals (*green*) to be unpaired. (D) RNA sequence and structural motifs recognized by the *S. cerevisiae* and *S. pombe* Ire1. (E) In vitro cleavage assay using an RNA hairpin derived from human *XBP1* mRNA 3' splice site, which is predicted to be a shared substrate for *S. cerevisiae* and *S. pombe* Ire1 KR. The calculated $k_{obs}$ is $16.7 \pm 2.3 \times 10^{-4}$ s$^{-1}$ for *S. cerevisiae* Ire1 KR and $38.9 \pm 4.0 \times 10^{-4}$ s$^{-1}$ for *S. pombe* Ire1 KR. (F) Illustrations of the *S. pombe* *BIP1* mRNA variants and (G) their uncleaved (*dark grey*) or total (*light grey*) mRNA fold change upon ER stress in *S. pombe* cells. Experiments were done in triplicates.

DOI: https://doi.org/10.7554/eLife.35388.006

The following figure supplements are available for figure 3:

**Figure supplement 1.** *S. pombe* Ire1 cleaves at UG|C positioned near the center of loops in RNA stem-loop structures.

DOI: https://doi.org/10.7554/eLife.35388.007

**Figure supplement 2.** Ire1 cleavage sites on *BIP1* mRNA variants.

DOI: https://doi.org/10.7554/eLife.35388.008

membered loop to a 7-membered one. The new construct ('*BIP1-HAC1* hybrid mRNA') contained the sequence AG|C, which is different from the *S. pombe* Ire1 sequence motif UG|C. In support of our model, *S. pombe* Ire1 failed to cleave the *BIP1-HAC1* hybrid mRNA upon UPR induction (*Figure 3G*, bar 3 and 4, *Figure 3—figure supplement 2A,B*). Next, we mutated the non-cognate A to a cognate U, such that the *S. pombe* Ire1 cleavage motif UG|C was restored. Indeed, the single nucleotide change restored the *S. pombe* Ire1 cleavage (*Figure 3G*, bar 5 and 6, *Figure 3—figure supplement 2C*). Taken together, we conclude that the Ire1 orthologs in *S. cerevisiae* and *S. pombe* have divergent substrate preferences.

## Engineering non-conventional mRNA splicing in *S. pombe*

While these results revealed the differences between the *S. cerevisiae* and *S. pombe* UPR at the step of Ire1 cleavage, it was not clear what determines the fates of the RNA cleavage fragments, that is RNA ligation in *S. cerevisiae* and RNA degradation in *S. pombe*. To address this question, we tested whether *S. pombe* cells contain functional mRNA ligation machinery. To this end, we expressed the *S. cerevisiae* *HAC1* mRNA-derived splicing reporter (*Figure 1—figure supplement 2A*) in Δ*ire1* *S. pombe* cells, bearing genomic copies of various Ire1 constructs. A chimeric Ire1 bearing the *S. pombe* cytosolic domain failed to splice the reporter mRNA, in agreement with our model (*Figure 4A*, lane 5 and 6). Interestingly, both Ire1 constructs bearing the *S. cerevisiae* cytosolic domains successfully spliced the reporter mRNA (*Figure 4A*, lanes 1–4). Thus, the *S. pombe* cells ligated the mRNA cleavage fragments as long as the correct RNA substrates were provided. This result suggested that features in the RNA substrates determine their fate post Ire1 cleavage.

Recently, we reported that in mammalian cells the *XBP1* mRNA actively participates in the splicing reaction. In particular, a conformational RNA rearrangement promotes *XBP1* mRNA intron ejection and exon ligation (*Peschek et al., 2015*). We wondered if this mechanism could be the factor that diverges the fates of the RNA cleavage fragments. To address this question, we aimed to synthetically create the Ire1-dependent non-conventional mRNA splicing reaction in *S. pombe* cells initiated by endogenous *S. pombe* Ire1. First, we identified the analogous RNA conformational rearrangement in *S. cerevisiae* *HAC1* mRNA. To obtain a *HAC1*-derived RNA splicing cassette optimized for *S. pombe* Ire1, we then engineered the two Ire1-cleavage sites at the splice junctions to match the *S. pombe* Ire1 UG|C motif and pruned the intron (originally 252 bp in *S. cerevisiae*) to the very residues predicted to be critical for the mRNA conformational rearrangement (30 bp). Finally, we inserted the *S. pombe*-optimized mRNA splicing cassette into *S. pombe* *BIP1* mRNA, replacing its endogenous Ire1 cleavage site (*Figure 4B*, *Figure 4—figure supplement 1*). Indeed, we found that the *BIP1* mRNA containing the synthetic splicing cassette was spliced in *S. pombe* upon induction of ER stress (*Figure 4C* lane 2). Sequencing of the lower band in *Figure 4C* (lane 2) verified the designed identity of the splicing product (*Figure 4D*). To show that insertion of the splicing cassette triggers mRNA splicing independent of particular flanking elements, we inserted the splicing cassette into the 3'UTR of another synthetic mRNA. In this case, we constructed a synthetic mRNA containing the 5' UTR of tubulin (*NDA2*) mRNA, the open reading frame of a *GAS2* mutant mRNA in which all of its RIDD cleavage sites were mutated, and the 3' UTR of *NDA2* mRNA with the inserted splicing cassette (*Figure 4E*). As for the mRNA described above, this construct was efficiently spliced

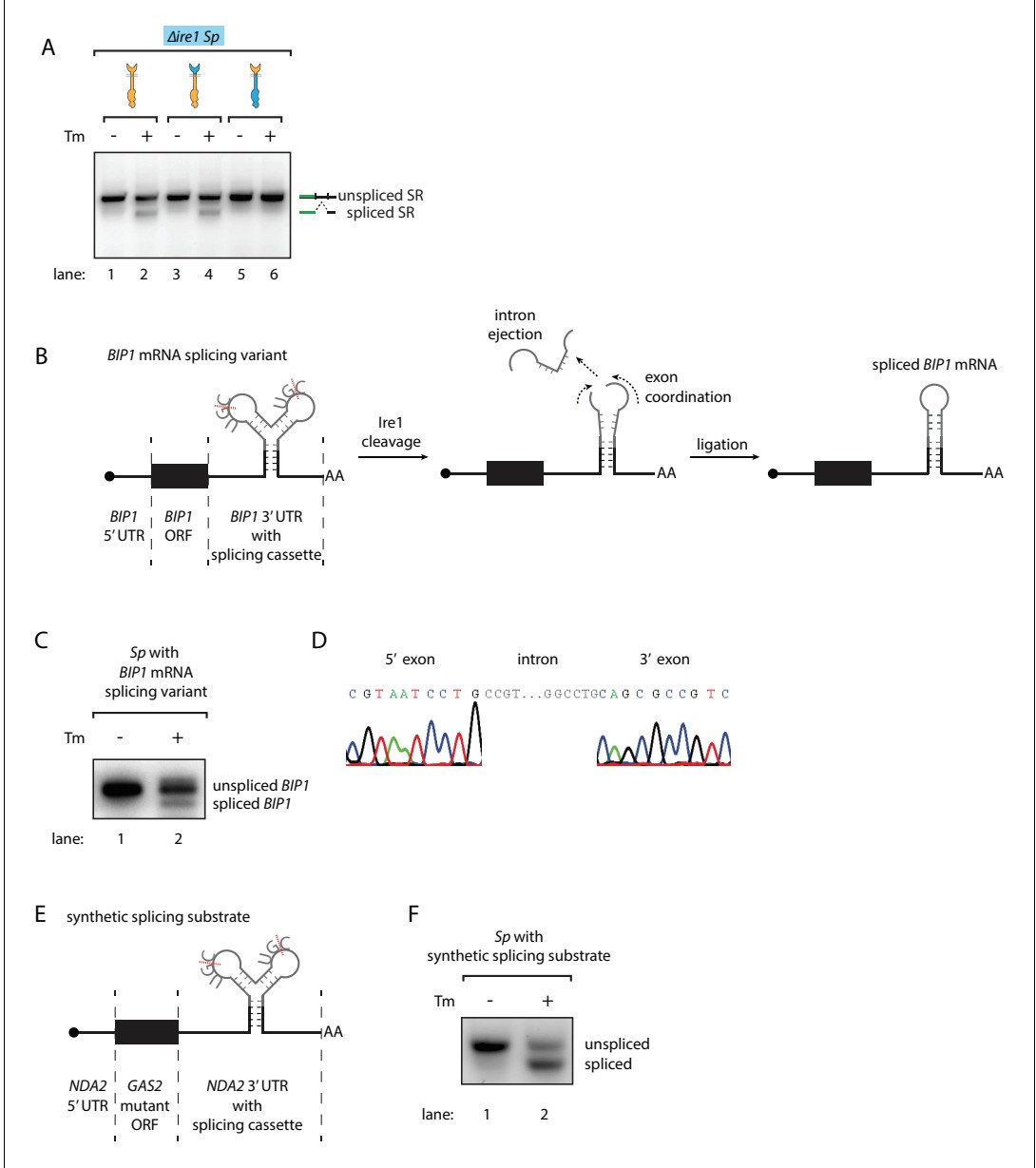

**Figure 4.** Engineering the Ire1-mediated non-conventional mRNA splicing in *S. pombe* cells. (**A**) Measuring the non-conventional mRNA splicing in *S. pombe* cells, which were transformed with the *S. cerevisiae HAC1* mRNA splicing reporter (SR) and the indicated Ire1 constructs. Cells were treated with 1 μg/ml Tm for 1 hr. (**B**) Illustration of the engineered *S. pombe BIP1* mRNA splicing variant. (**C**) Measuring the non-conventional mRNA splicing of the engineered *S. pombe BIP1* mRNA splicing variant. Experimental conditions are the same as those for *Figure 4A*. (**D**) Sequencing reads of the spliced *BIP1* mRNA. The schematic illustration (**E**) and the splicing assay (**F**) of the synthetic splicing substrate in *S. pombe*. Cells were treated with 1 μg/ml Tm for 1 hr.

DOI: https://doi.org/10.7554/eLife.35388.009

The following figure supplements are available for figure 4:

**Figure supplement 1.** The splicing cassette in the engineered *S. pombe BIP1* mRNA splicing variant.

DOI: https://doi.org/10.7554/eLife.35388.010

**Figure supplement 2.** The Ire1α cleavage sites on *XBP1* mRNA and RIDD targets.

DOI: https://doi.org/10.7554/eLife.35388.011

**Figure supplement 3.** The sequence alignment of the kinase/RNase domains of Ire1α, Ire1β, the *S. cerevisiae* Ire1 and the *S. pombe* Ire1.

DOI: https://doi.org/10.7554/eLife.35388.012

in *S. pombe* upon ER stress (*Figure 4F*). Thus, we conclude that the substrate RNA structure determines the fate of the RNA cleavage fragments.

## Discussion

'What I cannot create, I do not understand' (Richard Feynman). Inspired by this quote, we engineered *S. pombe* cells to carry out a non-conventional mRNA splicing reaction that does not occur in this species naturally. This feat was enabled by a combination of a detailed mechanistic characterization of the differences in Ire1-dependent mRNA processing between two yeast species reported in this paper and by functional insights gleaned previously from a characterization of the mammalian *XBP1* mRNA splicing mechanism (*Peschek et al., 2015*). Two main conclusions emerged from this study. First, the evolutionary divergence of Ire1's substrate specificity determines the set of mRNAs that are cleaved in either species. Second, features in the mRNAs determine the distinct fates of the severed mRNA fragments. *S. pombe* can be coopted to carry out the core non-conventional mRNA splicing reaction with fidelity, as long as it is provided with a spliceable RNA substrate that matches the specificity requirements of its endogenous Ire1. Albeit not optimized in evolution for efficiency of the reaction, no other components (such as the RNA ligase) need to be fundamentally specialized to mediate the core splicing reaction.

Ire1 orthologs in *S. cerevisiae* and *S. pombe* recognize their cognate mRNA substrates by discriminating both sequence and structural features. *S. pombe* Ire1 RNase specificity is more promiscuous, and has a broad substrate scope, in line with its role to initiate degradation of many ER-bound mRNAs to reduce the ER's protein folding burden. By contrast, *S. cerevisiae* Ire1's RNase specificity is very stringent and specialized with *HAC1* mRNA being as its only substrate in the cell (*Niwa et al., 2005*), in line with its role to produce a single transcription activator to drive UPR target genes. In mammals, two paralogs of Ire1 are expressed in a tissue-specific manner (*Tsuru et al., 2013*; *Bertolotti et al., 2001*). Ire1α, which performs both *XBP1* mRNA splicing and RIDD, recognizes a similar but longer RNA sequence motif, CUG|CAG, displayed in stem-loop structures (*Maurel et al., 2014*). Interestingly, the loop sizes that Ire1α recognizes differ between the *XBP1* mRNA cleavage sites and RIDD cleavage sites, with the two cleavage sites on *XBP1* mRNA to be conserved 7-mer loops (*Hooks and Griffiths-Jones, 2011*), while the cleavage sites on RIDD substrates vary in range from 9-mers to 5-mers (*Figure 4—figure supplement 2*). This suggests that mammalian Ire1α may display two distinct modes of RNase activity—a more stringent mode of RNase activity observed on 7-mer stem-loop RNAs (*XBP1* mRNA and *BLOC1S* mRNA) and a more promiscuous mode of RNase activity on RNA substrates with variable loop sizes. As shown here, these two modes have been cleanly separated in evolution between *S. cerevisiae* and *S. pombe* Ire1. Hence it seems plausible that mammalian Ire1α may switch selectively into one or the other state, perhaps in response to the timing of UPR activation or certain physiological conditions, which could be reflected in particular oligomeric states, post-translational modifications, or other effectors yet to be discovered.

Ire1α is more efficient in *XBP1* mRNA splicing, while Ire1β prefers to cleave ribosomal RNA (*Iwawaki et al., 2001*; *Nakamura et al., 2011*; *Imagawa et al., 2008*). Pairwise sequence alignment did not reveal an obvious similarity signatures that would distinguish the Ire1 species performing for RIDD, that is, *S. pombe* Ire1 and Ire1β, from those engaged in mRNA splicing, that is, *S. cerevisiae* Ire1 and Ire1α (*Figure 4—figure supplement 3A,B*). On the substrate side, two Ire1β cleavage sites both located on rRNA, have been mapped to date. They share a common sequence of G|C at the cleavage site. Previous studies indicated that differences in Ire1α's and Ire1β's RNase domains lead to their functional distinction (*Imagawa et al., 2008*). We did not observed cleavage of rRNA by *S. pombe* Ire1 (*Figure 1—figure supplement 1A,B*); mammalian Ire1β's activity to do so may therefore reflect a specialization that is not generalizable to all Ire1 RNases that perform RIDD. Therefore, modulating Ire1's RNase specificity to regulate its mode of action emerges as a general theme for different species, as well as for different tissues within the same species.

In *S. cerevisiae* cells, apart from the *HAC1* mRNA, other mRNAs contain the *S. cerevisiae* Ire1 cleavage motif, yet are not cleaved (*Niwa et al., 2005*). This is explained by spatial coordination. *HAC1* mRNA is targeted to Ire1 upon stress, utilizing a specific signal in the *HAC1* mRNA 3'UTR (*Aragón et al., 2009*), conferring exquisite specificity that renders *HAC1* mRNA the sole substrate of the reaction. Although two other Ire1 substrate RNAs have been reported (*Tam et al., 2014*), we

have not been able to reproduce this result. By contrast to the dedicated mRNA targeting in *S. cerevisiae*, *S. pombe* and mammalian cells target *XBP1* mRNA and most RIDD substrates to the ER via the signal recognition particle (SRP) pathway (*Hollien and Weissman, 2006*; *Hollien et al., 2009*; *Yanagitani et al., 2009*; *Yanagitani et al., 2011*; *Plumb et al., 2015*). In this way, *S. pombe* and mammalian Ire1 efficiently sample through substrate RNAs at ER periphery, which they cleave more promiscuously. We presume that in our experimental set-up, the chimeric *BIP1*-mRNA containing the splicing cassette would highjack the SRP-mediated targeting route initiated by the BiP1 signal sequence, as we previously showed for other RIDD substrate mRNAs (*Kimmig et al., 2012*).

After Ire1 cleavage, RNA fragments are ligated or degraded, depending on the substrate RNA structures. According to this notion, in mammalian cells where splicing and RIDD co-exist, we predict that *XBP1* mRNA splicing and RIDD substrate degradation are separated post Ire1 cleavage. The two cleavage sites on the *XBP1* mRNA are coordinated by a zipper-like RNA structure, which enable the exons to be held in juxtaposition and ligated by the cytosolic tRNA ligase (*Peschek et al., 2015*; *Sidrauski et al., 1996*; *Jurkin et al., 2014*; *Lu et al., 2014*; *Kosmaczewski et al., 2014*). By contrast, cleavage sites on the RIDD substrates lack such coordination and the cleavage fragments are further degraded.

Our study revealed that Ire1's RNase specificity and its RNA substrate structure separate Ire1's modes of action, opening the door to identify residues that shape Ire1's RNase specificity. In this way, it should become possible to design metazoan Ire1s that favor mRNA splicing over RIDD, and *vice versa*, enabling us to discriminate the biological significance of the two Ire1 functional outputs separately in physiological and pathological contexts.

# Materials and methods

## Strains, plasmids and growth conditions

Standard *S. cerevisiae* and *S. pombe* genome editing and growth conditions were used (*Moreno et al., 1991*; *Guthrie and Fink, 2002*). Strains used in this study are listed in the *Table 1*. Specifically, all Ire1 constructs have a 3x FLAG-tag in their lumenal domains replacing an unstructured region (255–274 in *S. pombe* and 267–286 in *S. pombe*). *S. cerevisiae* Ire1 domain boundaries were previously described (*Rubio et al., 2011*), *S. pombe* Ire1 domains were determined by sequence alignment with *S. cerevisiae* Ire1. Specifically, the lumenal domain is 1–526 for *S. cerevisiae* and 1–507 for *S. pombe*. The transmembrane/cytosolic linker is 527–672 for *S. cerevisiae* and 508–651 for *S. pombe*. Kinase/RNase is 673–1115 for *S. cerevisiae* and 652–1073 for *S. pombe*. Ire1 constructs were integrated into the *HO* locus in *S. cerevisiae* (backbone plasmid: HO-Poly-KanMX4-HO) and *Leu* locus in *S. pombe* (backbone plasmid: pJK148). *S. pombe BIP1* variants were integrated at the *BIP1* locus through homologous recombination and uracil selection. The mCherry-tagged Ire1 constructs and the splicing reporter were previously described (*Aragón et al., 2009*).

## Growth assay

Serial dilutions of *S. cerevisiae* or *S. pombe* cells were spotted onto YPD plates with 0.1 µg/ml tunicamycin (for *S. cerevisiae*) or YE5S plates with 0.05 µg/ml tunicamycin (for *S. pombe*). Plates were photographed after incubating at 30°C for 4 days.

## Immunoblotting

For both *S. cerevisiae* and *S. pombe* cells, total protein was isolated from yeast cultures growing at exponential phase by vortexing with glass beads in 8 M urea, 50 mM Hepes, pH 7.4, and 1% sodium dodecylsulfate (SDS). Samples were boiled and then centrifuged at 16,000 x g for 10 min. A sample containing 20 µg total protein was separated using electrophoresis and then transferred to nitrocellulose. The 3xFLAG-tagged Ire1 was probed with monoclonal anti-FLAG antibody (Sigma F3165).

## qPCR assays

Total RNA was purified from yeast cultures using phenol extraction (*Köhrer and Domdey, 1991*). RNA samples were resuspended in RNase-free water and quantified by spectrophotometry. cDNA was synthesized by reverse transcription using random hexamer DNA primers (Thermo Fisher Scientific), SuperScript II Reverse Transcriptase (Thermo Fisher Scientific) and 1 µg total RNA as described

**Table 1.** Yeast strains used in this study.

All strains are derived from WL001 and WL002. All Ire1 constructs listed below are 3x FLAG-tagged within their lumenal domains.

| Strain | Species | Description |
| --- | --- | --- |
| yWL001 | *Sc* | WT, mat A, *leu2-3,112 TRP1 can1-100 ura3-1 ADE2 his3-11,15* |
| yWL002 | *Sc* | *ire1Δ::NAT^R* |
| yWL003 | *Sc* | *ire1Δ::NAT^R, HO::Sp IRE1* |
| yWL004 | *Sc* | *ire1Δ::NAT^R, HO::Sp*Lum*Sc*Cyto *IRE1* |
| yWL005 | *Sc* | *ire1Δ::NAT^R, HO::Sc*Lum*Sp*Cyto *IRE1* |
| yWL006 | *Sc* | *ire1Δ::NAT^R, HO::Sc IRE1* |
| yWL007 | *Sc* | *ire1Δ::NAT^R, HO::Sc*Lum/transmembrane/linker*Sp*KR *IRE1* |
| yWL008 | *Sc* | WT, *leu2::5'hac1-gfp-3'hac1* |
| yWL009 | *Sc* | *ire1Δ::NAT^R, leu2::5'hac1-gfp-3'hac1* |
| yWL010 | *Sc* | *ire1Δ::NAT^R, leu2::5'hac1-gfp-3'hac1, HO::Sp IRE1* |
| yWL011 | *Sc* | *ire1Δ::NAT^R, leu2::5'hac1-gfp-3'hac1, HO:: Sp*Lum*Sc*Cyto *IRE1* |
| yWL012 | *Sc* | *ire1Δ::NAT^R, leu2::5'hac1-gfp-3'hac1, HO:: Sc*Lum*Sp*Cyto *IRE1* |
| yWL013 | *Sc* | *ire1Δ::NAT^R, leu2::5'hac1-gfp-3'hac1, HO::Sc IRE1* |
| yWL014 | *Sc* | *leu2::5'hac1-gfp-3'hac1, his3::pTdh3-mCherry* |
| yWL015 | *Sp* | WT, mat h^+, *ade6-M210, ura4-D18, leu1-32* |
| yWL016 | *Sp* | *ire1Δ::KAN^R* |
| yWL017 | *Sp* | *ire1Δ::KAN^R, leu1::Sp IRE1* |
| yWL018 | *Sp* | *ire1Δ::KAN^R, leu1::Sp*Lum*Sc*Cyto *IRE1* |
| yWL019 | *Sp* | *ire1Δ::KAN^R, leu1::Sc*Lum*Sp*Cyto *IRE1* |
| yWL020 | *Sp* | *ire1Δ::KAN^R, leu1::Sc IRE1* |
| yWL021 | *Sp* | *ire1Δ::KAN^R, leu1::Sp*Lum*Sc*Cyto *IRE1, ura4::5'hac1-gfp-3'hac1* |
| yWL022 | *Sp* | *ire1Δ::KAN^R, leu1::Sc*Lum*Sp*Cyto *IRE1, ura4::5'hac1-gfp-3'hac1* |
| yWL023 | *Sp* | *ire1Δ::KAN^R, leu1::Sc IRE1, ura4::5'hac1-gfp-3'hac1* |
| yWL024 | *Sp* | *bip1::bip1-hac1* hybrid |
| yWL025 | *Sp* | *bip1::bip1-hac1* hybrid A->U |
| yWL026 | *Sp* | *bip1::bip1* splicing variant |
| yWL027 | *Sp* | *ura4::synthetic* splicing substrate |

DOI: https://doi.org/10.7554/eLife.35388.013

previously (*Kimmig et al., 2012*). 1% of the cDNAs was employed for qPCR reactions using SYBR green qPCR kit (Bio-Rad). qPCR was performed in triplicates using CFX96 Touch Real-Time PCR Detection System (Bio-rad). qPCR primers are listed in *Table 2*. mRNA levels were normalized to *NDA2* mRNA in *S. pombe*.

## In vivo mRNA splicing assay

cDNA was synthesized the same way as described in the qPCR section. Then we used Phusion High-Fidelity PCR Kit (NEB) and performed PCR with cDNA and a set of primers across the splice junction. For *HAC1* mRNA, the forward primer was ATGGAAATGACTGATTTTGAACTAACTAGTAATTCG. The reverse primer was TCATGAAGTGATGAAGAAATCATTCAATTCAAATG. The PCR was performed for 26 cycles with annealing temperature of 51.5°C and extension time of 30 s. For the *S. pombe BIP1* mRNA containing the splicing cassette, the forward primer was GAATCGTGACTCTATAGCCATTAACA. The reverse primer was CAATTATTGTCAGTTCCACAAAGC. The PCR was performed for 36 cycles with annealing temperature of 63.4°C and extension time of 15 s. For *S. cerevisiae HAC1* mRNA derived splicing reporter expressed in *S. pombe* cells, the forward primer was GAACTACAAGACACGTGCTGAAG. The reverse primer was GATGAAGAAATCATTCAATTCAAA

**Table 2.** qPCR primers used in this study.

| qPCR primers description | Sequence |
| --- | --- |
| uncleaved *Sp BiP1* mRNA forward primer | GAATCGTGACTCTATAGCCATTAACA |
| uncleaved *Sp BiP1* mRNA reverse primer | CAATTATTGTCAGTTCCACAAAGC |
| total *Sp BiP1* mRNA forward primer | TGGTAAGGTTGATCCCGAAG |
| total *Sp BiP1* mRNA reverse primer | CATCGAGTTTTTGACGCTGA |
| *Sp GAS2* mRNA forward primer | GTTGTCAACAATGCCTCGAA |
| *Sp GAS2* mRNA reverse primer | CGGTCTCAGAGTTGGTGTCA |
| *Sp NDA2* mRNA forward primer | TCCATGAATCCAACAGCGTA |
| *Sp NDA2* mRNA reverse primer | CTAGTAACGGCAGCCTGGAC |

DOI: https://doi.org/10.7554/eLife.35388.014

TG. The PCR was performed for 60 cycles with annealing temperature of 63.2°C and extension time of 20 s. For the synthetic splicing substrate in *S. pombe*, the forward primer was CTCATTTAGATTAGCAATTCAAATG. The reverse primer was GATTAGATCAACAATTCAAATGATC. The PCR was performed for 40 cycles with annealing temperature of 59.7°C and extension time of 20 s.

## Recombinant protein purification

*S. cerevisiae* Ire1 kinase/RNase was purified as previously described (*Korennykh et al., 2009*). Details of the *S. pombe* Ire1 kinase/RNase purification will be described in a separate paper. Briefly, *S. pombe* Ire1 kinase/RNase was N-terminally fused with Glutathione S-transferase (GST) tag through a linker containing Human Rhinovirus (HRV) 3C protease cleavage site, and was regulated by T7 promoter. This *S. pombe* Ire1 kinase/RNase expression cassette was transformed into *E. coli* cells. 16 hr after transformation, we mixed and collected all the colonies on the transformation plates by scraping them off from the agar plate into 50 ml of LB medium. After 3 hr incubation at 37°C, the sample was diluted to 12 l of LB medium and further incubated at 37°C until optical density reached 1. Protein expression was induced by adding 0.5 mM IPTG. Then, the culture was incubated at 25°C for 4 hr before we pelleted the cells by centrifugation. Cells were resuspended in GST binding buffer (50 mM Tris/HCl pH 7.5, 500 mM NaCl, 2 mM Mg(OAc)$_2$, 2 mM DTT, 10% Glycerol) and homogenized using high-pressure homogenizer (EmulsiFlex, Avestin). The cell lysate was applied to GST-affinity column and eluted with GST elusion buffer (50 mM Tris/HCl pH 7.5, 200 mM NaCl, 2 mM Mg(OAc)$_2$, 2 mM DTT, 10% Glycerol, 10 mM glutathione). The column elution was treated with GST-tagged HRV 3C protease (PreScission Protease, GE Health). At the same time, the sample was dialyzed to remove glutathione in the elution buffer. Next, the sample was further purified through negative chromatography by passing through a GST-affinity column (to remove free GST and residue GST-fused Ire1 kinase/RNase) and an anion exchange column (to remove contaminating nucleic acids). Finally, the sample was subject to gel filtration, concentrated to about 14 μM in storage buffer (50 mM Tris/HCl pH 7.5, 200 mM NaCl, 2 mM Mg(OAc)$_2$, 2 mM TCEP, 10% Glycerol), and flash frozen in liquid nitrogen. The final purity, as well as purity at intermediate steps, was assessed by SDS-PAGE using Coomassie blue staining.

## In vitro RNA cleavage assays

Short RNA oligos were purchased from Dharmacon, Inc. RNA oligos were gel extracted, acetone precipitated and resuspended in RNase-free water. Then, oligos were 5′ end radio-labeled with γ-[$^{32}$P]-ATP (Perkin Elmer) using T4 polynucleotide kinase (NEB) and cleaned using ssDNA/RNA Clean and Concentrator kit (Zymo Research D7010). To fold the RNA oligos, we heated the RNA oligos to 90°C for 3 min and slowly cooled them down at a rate of 1°C per minute until the temperature reached 10°C. In the Ire1 cleavage assays, the reaction samples contained 12.5 μM of *S. cerevisiae* or *S. pombe* Ire1 kinase/RNase. The cleavage reaction was performed at 30°C in reaction buffer (50 mM Tris/HCl pH 7.5, 200 mM NaCl, 2 mM Mg(OAc)$_2$, 2 mM TCEP, 10% Glycerol). At each time point, an aliquot of 0.75 μl was transferred to 5 μl STOP buffer (10 M urea, 0.1% SDS, 1 mM EDTA, 0.05% xylene cyanol, 0.05% bromophenol blue). RNAs were separated using denaturing 15% urea-

PAGE gels (run at 100 V for 90 min). Gels were imaged with a Phosphorimager (Typhoon FLA 9500, GE Health) and the band intensities were quantified using imageJ. The cleaved portion was calculated as the cleaved band intensity divided by the sum of the cleaved band and uncleaved band intensities. The $k_{obs}$ were obtained by fitting the data to first-order ('one-phase') decay equation using Prism. For the cleavage reactions that less than 10% of the substrates were cleaved, because the substrate concentration was approximately constant, the cleavage dynamics was fitted to a linear equation to obtain $k_{obs}$. The sequence of hairpin RNA substrate derived from Ire1 cleavage site on *S. pombe BIP1* mRNA is CGCGAGAUAACUGGUGCUUUGUUAUCUCGCG.

The sequence of hairpin RNA substrate derived from Ire1 cleavage site on *S. pombe SPAC4G9.15* mRNA is CCACCACCGAGUAUGCUACUCGGUGGUGG.

The sequence of hairpin RNA substrate derived from *S. cerevisiae HAC1* mRNA 3' splice site is GCGCGGACUGUCCGAAGCGCAGUCCGCGC

The sequence of hairpin RNA substrate derived from *XBP1* mRNA 3' splice site is UGCACCUCUGCAGCAGGUGCA.

## Automated flow cytometry

Measuring *S. cerevisiae* UPR dynamics using automated flow cytometry was previously described in detail (*Zuleta et al., 2014*). Briefly, we co-cultured two *S. cerevisiae* strains, a strain of interest and a control strain. The control strain contained a constitutively expressed mCherry reporter. The signal from the control strain was computationally separate based on its high mCherry level. In an 11.5 hr time course at 30°C, a data point was taken every 20 min. 1.5 hr after inoculation, DMSO (as control) or 0.25 µg/ml, 0.5 µg/ml, 1 µg/ml, 2 µg/ml of Tm were added. Splicing dynamics were monitored for another 10 hr. The GFP signal was normalized to the signal at time zero.

## Probing in vivo mRNA structure in *S. pombe* cells

A culture of 15 ml *S. pombe* cells, which were exponentially growing at 30°C, was treated with 400 µl of DMS for 3.5 min. DMS was then quenched by adding 30 ml of solution containing 30% β-mercaptoethanol and 25% isoamyl alcohol. Then, cells were pelleted by centrifugation at 4°C, and washed with 15 ml 30% β-mercaptoethanol. Total RNA was extracted using phenol extraction. Poly (A)+ mRNAs were isolated using poly(A)+ Dynabeads (Invitrogen). The sequencing library was generated and sequenced, and the DMS modifications were computed as previously described (Rouskin et al.).

## mRNA secondary structure prediction

Near the Ire1 cleavage sites, we first identified the most highly reactive base and set its DMS modification signal as 1. Then, the DMS modification signal raw data for other bases was normalized proportionally. Finally, we put a 38-base-pair RNA sequence (19 base pair upstream and downstream from the Ire1 cleavage site) into the RNA secondary structure prediction program mfold (*Zuker, 2003*). Bases with normalized DMS modification signals >0.2 were forced to be single stranded to constrain the RNA folding prediction.

## Additional information

### Funding

| Funder | Grant reference number | Author |
| --- | --- | --- |
| Howard Hughes Medical Institute | | Jonathan S Weissman Peter Walter |
| Human Frontier Science Program | | Jirka Peschek Philipp Kimmig |
| National Science Foundation | | Meghan Zubradt |
| Genentech Foundation | | Meghan Zubradt |
| Center for RNA Systems Biology | | Jonathan S Weissman |

| University of California, San Francisco | UCSF-Zaffaroni Fellowship | Weihan Li |

The funders had no role in study design, data collection and interpretation, or the decision to submit the work for publication.

## Author contributions
Weihan Li, Conceptualization, Data curation, Formal analysis, Validation, Investigation, Visualization, Methodology, Writing—original draft, Writing—review and editing; Voytek Okreglak, Conceptualization, Data curation, Supervision, Methodology, Writing—review and editing; Jirka Peschek, Supervision, Methodology, Writing—review and editing; Philipp Kimmig, Conceptualization, Supervision, Methodology, Writing—review and editing; Meghan Zubradt, Data curation, Formal analysis, Investigation, Methodology, Writing—review and editing; Jonathan S Weissman, Resources, Supervision, Funding acquisition, Writing—review and editing; Peter Walter, Conceptualization, Resources, Supervision, Funding acquisition, Writing—review and editing

## Author ORCIDs
Weihan Li (iD) http://orcid.org/0000-0003-4718-1884
Jirka Peschek (iD) http://orcid.org/0000-0001-8158-9301
Jonathan S Weissman (iD) http://orcid.org/0000-0003-2445-670X
Peter Walter (iD) http://orcid.org/0000-0002-6849-708X

## Decision letter and Author response
Decision letter https://doi.org/10.7554/eLife.35388.019
Author response https://doi.org/10.7554/eLife.35388.020

# Additional files

## Supplementary files
• Transparent reporting form
DOI: https://doi.org/10.7554/eLife.35388.015

## Data availability
DMS-seq data have been deposited to Data Dryad.

The following dataset was generated:

| Author(s) | Year | Dataset title | Dataset URL | Database, license, and accessibility information |
|---|---|---|---|---|
| Li W, Okreglak V, Peschek J, Kimmig P, Zubradt M, Weissman J, Walter P | 2018 | Data from: Engineering ER-stress dependent non-conventional mRNA splicing | http://dx.doi.org/10.5061/dryad.s95f1 | Available at Dryad Digital Repository under a CC0 Public Domain Dedication |

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
