## [Decision Letter]

Thank you for submitting your article "Engineering ER-stress dependent non-conventional mRNA splicing" for consideration by *eLife*. Your article has been reviewed by two peer reviewers, and the evaluation has been overseen by a Reviewing Editor and Vivek Malhotra as the Senior Editor. The reviewers have opted to remain anonymous.

The reviewers have discussed the reviews with one another and the Reviewing Editor has drafted this decision to help you prepare a revised submission.

Summary:

In this study, Li et al., investigate the ER-stress sensor and effector activities of Ire1 in *S. pombe* and *S. cerevisiae*. They exploit the fact that the Ire1 orthologs in the two disparate yeast species have different functions; *HAC1* exhibits non-conventional mRNA splicing in *S. cerevisiae*, while in *S. pombe* RIDD effects regulated Ire1-dependent decay, to understand the specificities of these two mechanisms, that coexist in Metazoans. The authors show that the divergence in the two Ire1 functions stems mainly from their respective kinase/RNase domain, that cleaves different substrate mRNAs upon stress by recognizing distinct RNA sequence and structure. Finally, they engineer an *S. pombe* mRNA target to be spliced after Ire1 cleavage, showing that the "fate" of the mRNA after cleavage by Ire1 (non-conventional splicing or degradation) is an intrinsic feature of the mRNA rather than the Ire1 function.

Essential revisions:

The two referees agree that the experiments presented in this paper are overall clean and well-presented. However, they thought that the paper fails to convey the relevance and novelty of the study. In Metazoans where both Ire1 functions exist, it was previously shown that the Ire1α and Ire1β differential functions can be explained by their RNAse domains (Imagawa et al., 2008). *S. cerevisiae* Ire1 was also shown to cleave mammalian RIDD targets in vitro (Tam et al., 2014). It was published that RNAse domains are responsible for different substrates (Imagawa et al., 2008). Also, *S. cerevisiae* Ire1 was shown to cleave mammalian RIDD targets in vitro (Tam et al., 2014).

The significance of the paper would be strongly strengthened if the authors could extend their conclusions to the mammalian Ire1 models. They should discuss the sequence and function variation of the mammalian Ire1 (or Ire1α and Ire1β) as compared to *S. pombe* and *S. cerevisiae* Ire1. Importantly, can they predict mRNAs structures in known targets of the mammalian Ire1 similar to the structures identified here (especially in RIDD targets compared to *S.pombe, XBP1/HAC1* structures being already identified)? It is also important to get a grasp of how is the engineering of unconventional splicing going to "help discriminate Ire1 function in physiological/pathological contexts".

Relevant to the last point, if a main advance from the paper is that the fate of the mRNA after cleavage by Ire1 is determined by the mRNA features, the results in Figure 4 are rather weak to support this conclusion. In A, although there is splicing of the *S. cerevisiae HAC1* in the presence of *S. cerevisiae* Ire1 in *S. pombe*, it seems to be lower as compared to *HAC1* in *S. cerevisiae* (Figure 1H and Figure 2B), suggesting that it does actually involve more than just the correct RNA substrate. Furthermore, are the results obtained with the engineered BIP1 mRNA reproducible with other targets? Does the produced mRNA have a biological function or how is its function altered?

---

## [Author Response]

Essential revisions:The two referees agree that the experiments presented in this paper are overall clean and well-presented. However, they thought that the paper fails to convey the relevance and novelty of the study. In Metazoans where both Ire1 functions exist, it was previously shown that the Ire1α and Ire1β differential functions can be explained by their RNAse domains (Imagawa et al., 2008). S. cerevisiae Ire1 was also shown to cleave mammalian RIDD targets in vitro (Tam et al., 2014). It was published that RNAse domains are responsible for different substrates (Imagawa et al., 2008). Also, S. cerevisiae Ire1 was shown to cleave mammalian RIDD targets in vitro (Tam et al., 2014).

Ire1β cleaves ribosomal RNA, along with a few mRNAs. Ire1β's RNase domain determines its rRNA cleavage ability (Imagawa et al. 2008). However, we didn't observe rRNA cleavage in *S. pombe* cells (new Figure 1—figure supplement 1). Thus, mammalian Ire1β’s rRNA cleavage activity may reflect a specialization that is not generalizable to all Ire1 RNases that perform RIDD.

In our study on the *S. cerevisiae* and *S. pombe* Ire1, we identified Ire1 recognition motifs, based on which we engineered non-conventional mRNA splicing in *S. pombe*.

Similar characterization for Ire1β is lacking. Our work thus provides a first template for understanding still to be gained in the mammalian system.

In regard to the observation that the *S. cerevisiae* Ire1 cleaving mammalian RIDD targets in vitro (Tam et al. 2014), we found that some mammalian RIDD targets, like *BLOC1S* mRNA (Figure 4—figure supplement 2), contain the *S. cerevisiae* Ire1 recognition motif. Therefore, these mRNAs are expectedly subject to cleavage by the *S. cerevisiae* Ire1.

The significance of the paper would be strongly strengthened if the authors could extend their conclusions to the mammalian Ire1 models. They should discuss the sequence and function variation of the mammalian Ire1 (or Ire1α and Ire1β) as compared to S. pombe and S. cerevisiae Ire1. Importantly, can they predict mRNAs structures in known targets of the mammalian Ire1 similar to the structures identified here (especially in RIDD targets compared to S.pombe, XBP1/HAC1 structures being already identified)?

We accept the reviewers’ suggestion and modified our Discussion section to include the following point:

Mammalian Ire1α performs both reactions, *XBP1* mRNA splicing and RIDD, yet RNA loop sizes differ between splicing and RIDD substrates: *XBP1* mRNA splicing sites are evolutionarily conserved as 7-mers while RIDD cleavage sites are flexible and rang from 9-mers to 5-mers (new Figure 4—figure supplement 2). Such loop size differentiations resemble the *HAC1* mRNA in *S. cerevisiae* (7mer) and RIDD substrates in *S. pombe* (flexible loop size). This comparison suggests that the feature of different loop size recognition can be extended from *S. cerevisiae* and *S. pombe* Ire1 to different function modes for mammalian Ire1α.

It is also important to get a grasp of how is the engineering of unconventional splicing going to "help discriminate Ire1 function in physiological/pathological contexts".

We have clarified the text. In particular we meant – and are now stating more clearly – that “after the molecular features that shape *S. cerevisiae* and *S. pombe* Ire1 RNase specificity are characterized, it will then be possible to engineer mammalian Ire1 constructs that either are more stringent and favor *XBP1* mRNA splicing, or more promiscuous and favor RIDD. Using such constructs, it will then become possible to separate Ire1’s disparate functional outputs and assess their respective roles in physiological/pathological contexts.”

Relevant to the last point, if a main advance from the paper is that the fate of the mRNA after cleavage by Ire1 is determined by the mRNA features, the results in Figure 4 are rather weak to support this conclusion. In A, although there is splicing of the S. cerevisiae HAC1 in the presence of S. cerevisiae Ire1 in S. pombe, it seems to be lower as compared to HAC1 in S. cerevisiae (Figure 1H and Figure 2B.), suggesting that it does actually involve more than just the correct RNA substrate.

The reviewers are correct. When both *HAC1* mRNA and *S. cerevisiae* Ire1 are expressed in *S. pombe*, the splicing efficiency is lower compared to the *HAC1* mRNA splicing efficiency in *S. cerevisiae* cells.

We modified our Discussion section to acknowledge the following two points:

1) The reduced efficiency of *HAC1* mRNA splicing in the heterologous context may result because other contributing components are not optimized in a species that does not normally carry out this reaction. For example, *HAC1* mRNA targeting to the ER and/or RNA ligation may occur suboptimally in *S. pombe* cells.

2) Given these caveats, it is amazing to find that the reaction can be reconstituted with fidelity (albeit less efficiently), as long as we provide the correct RNA substrate (Figure 4A, C, F).

Furthermore, are the results obtained with the engineered BIP1 mRNA reproducible with other targets?

We added new data and showed that the engineered non-conventional mRNA splicing in *S. pombe* works in other targets (Figure 4E, F).

Does the produced mRNA have a biological function or how is its function altered?

The spliced and unspliced *BIP1* mRNAs have the same coding sequence, the spliced *BIP1* mRNA is expected to have the same function as the un-spliced one. In this proof-of-principle experiment, the mRNA only served as a reporter tool.